# Efficient Large-scale Transformer Training via Random and Layerwise Token Dropping

## Abstract

Large-scale transformer models have become the de-facto architectures for various machine learning applications, e.g., CV and NLP. However, those large models also introduce prohibitive training costs. To mitigate this issue, we propose a novel random and layerwise token dropping method (random-LTD), which skips the computation of a subset of the input tokens at all middle layers. Particularly, random-LTD achieves considerable speedups and comparable accuracy as the standard training baseline. Compared to other token dropping methods, random-LTD does not require (1) any importance score-based metrics, (2) any special token treatment (e.g., [CLS]), and (3) many layers in full sequence length training except the first and the last layers. Besides, a new LayerToken learning rate schedule is proposed for pretraining problems that resolve the heavy tuning requirement for our proposed training mechanism. Finally, we demonstrate that random-LTD can be applied to broader applications, including GPT and BERT pretraining as well as ViT and GPT finetuning tasks. Our results show that random-LTD can save about 33.3% theoretical compute cost and 25.6% wall-clock training time while achieving similar zero-shot evaluations on GPT-3$_{1.3B}$ as compared to baseline.

## 1 Introduction

Large-scale transformers have been demonstrated to have supreme performance on natural language processing (Tenney et al., 2019; Radford et al., 2019; Raffel et al., 2019), computer vision (Dosovitskiy et al., 2020), and other applications (Gong et al., 2021; Guo et al., 2021). However, both the pretraining procedure and some downstream finetuning tasks (e.g., long document summary) are time-consuming and resource-hungry. Thus, there is a need to speed up the training and reduce the compute cost for large-scale transformer pretraining and finetuning.

Recently, Hou et al. (2022) adopt the token pruning/dropping/bypassing technique (Kim et al., 2021; Goyal et al., 2020; Kim & Cho, 2020) from BERT inference to BERT pretraining by skipping the compute of part of the input tokens at some middle layers. The results of (Hou et al., 2022) (referred to as TokenBypass) show that it can theoretically reduce the pretraining cost by 25% for both BERT$_{base}$ and BERT$_{large}$ without losing accuracy on finetuning tasks. Although achieving great speedup, TokenBypass (1) needs an import-score metric to determine the dropped tokens and special token treatment to keep important tokens (e.g., [CLS]), both of which require manual designs; (2) has to keep the first half layers and the last layer (in total, half of the depth) in full sequence length training, which limits its layer-bypassing ability. (3) solely focuses on BERT Masked-LM pretraining tasks and has not been applied to other tasks, e.g., causal-LM. In this work, we address those challenges and introduce our random and layerwise token-dropping method (random-LTD). In summary, *our contributions* are as follows:

- All tokens are treated equally without any special token treatment or import-score measurement, i.e., no manual design, and are dropped in a purely random manner. Meanwhile, instead of fully bypassing the dropped token for all middle layers (Hou et al., 2022), each layer in random-LTD drops tokens independently from the other layers. This helps the multi-head attention in the middle layers capture the dependency relation across different tokens suggested in (Vig & Belinkov, 2019).
- random-LTD applies token dropping at all middle layers except the very first and last layers, which further reduces manual design and increases training efficiency. We also propose a new monotonic sequence length growth method as training evolves to (1) reduce the gradient noise introduced

by random-LTD for better convergence and (2) close the training and inference (autoregressive generation) gap, since random-LTD breaks the autoregressive manner in middle layers during training, for GPT models.

- To reduce the tuning effort for the newly proposed training procedure, we introduce a new LayerToken learning rate schedule, which scales the learning rate based on the sum of consumed tokens of each layer for pretraining tasks.[1] We show its superb performance for random-LTD on GPT/BERT pretraining compared to the standard iteration-based learning rate schedule.
- We extensively test random-LTD on both pretraining tasks, including GPT and BERT pretraining, and finetuning tasks, including causal-LM finetuning for GPT and image classification for ViT. For all tasks, random-LTD achieves similar accuracy as the original baseline method with up to 33.3% theoretical cost saving and up to 25.6% wall-clock time saving.
- Finally, we show that random-LTD has a potential regularization effect, which can be used for both pretraining and finetuning problems.

## 2 BACKGROUND

Transformer (Vaswani et al., 2017) architecture is a stack of transformer layers, each of which has two main ingredients, i.e., the multi-head attention (MHA) and the feed-forward connection network (FFC). Suppose the transformer has $l$ layers denoted as $L_1, \ldots, L_l$. Let $X_i \in \mathbb{R}^{s \times d}$ be the output tensor of $i-$th transformer layer, and $x_0$ the input (after embedding) of the transformer. Here $s$ is the sequence length and $d$ is the hidden dimension.

Token dropping (or token bypassing/pruning) (Kim et al., 2021; Goyal et al., 2020; Kim & Cho, 2020; Press et al., 2021; Wang et al., 2021) was originally proposed for BERT inference to reduce the computational overhead. In this case, if a token $i$ ($X_{j,i}$) is decided to be dropped at layer $j$ ($L_j$), the compute cost of this token through all remaining layers ($L_k$ where $k > j$) is eliminated. As such, the sequence length $s_i$ of the $i$-th layer's input $X_{i-1}$ will be a non-increasing array, i.e., $s_0 \geq s_1 \ldots \geq s_l$. However, such a configuration has been shown instability for adaptive token-dropping inference (Kim & Cho, 2020). Therefore, Kim & Cho (2020) utilize the sandwich rule and distillation from (Yu & Huang, 2019) to stabilize training and boost accuracy. But these two methods also significantly increase the training cost. Thus, such techniques cannot be applied to speed up the pretraining procedure. Recently, Hou et al. (2022) extended token dropping from inference to BERT pretraining (referred to as TokenBypass). Hou et al. (2022) use several importance scores/metrics to determine the dropped tokens, e.g., cumulative loss and frequency of each token. To overcome the training instability issue, the authors proposed two main mechanisms: (1) the sandwich token dropping rule, where the first (layer 1 to $i$) and the last few layers (layer $L_{l-j}$ to $L_l$) of the BERT capture all tokens (i.e., no token dropping) and the middle layers bypass $s' \leq s$ tokens from $L_i$ to $L_{l-j}$. Particularly, the authors (only) test on the encoder transformer (12-layer BERT$_{\text{base}}$ and 24-layer BERT$_{\text{large}}$), and let $i = l/2 - 1, j = 1, s' = s/2$. (2) special token treatment, where special tokens (e.g., [MASK], [CLS], [SEP]) are never dropped.

Compared to TokenBypass from (Hou et al., 2022), our random-LTD (1) does not require importance score metric, special token treatment, or the sandwich token dropping rule, which dramatically reduces the manual design effort; (2) has been broadly tested on pretraining tasks, including GPT and BERT, as well as finetuning tasks, including ViT classification and GPT causal-LM. Meanwhile, we found out that directly applying TokenBypass to causal-LM leads to severe accuracy degradation. Please see the detailed description of random-LTD in Section 3 and our extensive evaluation in Section 4 and 5. We also include a thorough discussion of other efficient training methods in Appendix A.

## 3 METHODOLOGY

### 3.1 RANDOM AND LAYERWISE TOKEN DROPPING METHOD

**Layerwise Token Dropping Mechanism.** As pointed out in Section 2, existing inference and training token dropping methods either permanently drop tokens from the compute graph at intermediate layers, or at least make some tokens fully skip a consecutive series of middle layers. However, several works (Vig & Belinkov, 2019; Michel et al., 2019; Voita et al., 2019) have shown that MHA focuses

---

[1]Note that the numbers of consumed tokens for different layers are different.

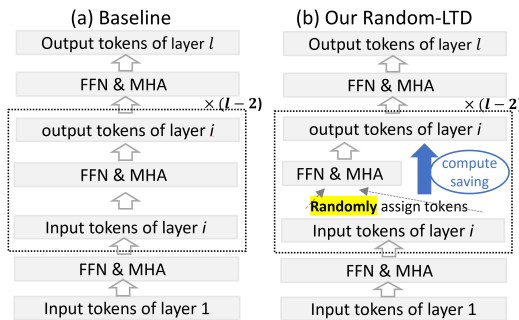

Figure 1: Transformer layers for baseline and random-LTD training. The dash-line box is repeated by $l - 2$ times (see more in Fig. B.1).

```
1  if meth == "baseline":
2      hs = Layer(hs)
3  if meth == "random-LTD":
4      k_hs, d_hs = gather(hs)
5      k_hs = Layer(k_hs, d_hs)
6      hs = combine(k_hs, d_hs)
```

Figure 2: random-LTD only requires a few lines of code. Here hs, $k_{hs}$, and $d_{hs}$ means the full input, kept input, and dropped input. "gather", "Layer", "combine" means the functions for random selection, transformer layer, and token combination.

on different tokens at different layer depths and the attention map aligns with the dependency relation most strongly in the middle of transformer architectures. Therefore, TokenBypass used in Hou et al. (2022), i.e., fully skipping middle layers, may hinder the learnability/generalization of the architecture during pretraining/inference. We conjecture that this might be why multiple first/last layers need to be kept and the special token treatment is needed in (Hou et al., 2022). To further verify if this fully skipping middle layer mechanism (Hou et al., 2022) causes any learnability issue, we apply TokenBypass on GPT finetuning tasks and observe much lower performance as compared to baseline. See more details in Section 5.1.

In order to overcome this problem, we now propose a layerwise token dropping (LTD) mechanism. Instead of fully bypassing dropped tokens over all middle layers, each transformer layer independently drops/retains its own set of tokens. In more detail, recall that the input of $(i + 1)$-th layer ($L_{i+1}$) is $X_i \in \mathbb{R}^{s \times d}$. Denote the dropped token index as $J_i = \{j_1, j_2, ..., j_{a_i}\}$ and the kept token index as $K_i = \{k_1, ..., k_{b_i}\}$ such that $a_i + b_i = s$. We have $J_i \cup K_i = \{1, 2, 3..., s\}$ and $J_i \cap K_i = \emptyset$ for each layer. Meanwhile, for any two different layers $L_{i_1}$ and $L_{i_2}$, $J_{i_1}$ and $J_{i_2}$ are independent, though the dropped ratios are the same. With this layerwise mechanism, each token rarely bypasses all middle layers. Thus, its dependency on other tokens can be captured by MHA.

**Random Token Dropping.** Various important score-based metrics are used to determine the token dropping criterion. Most of them can be categorized in two ways: attention score related metrics or loss/frequency-based metrics. However, both of them introduce challenges that make LTD less practical. Particularly, for attention score-based metrics, the compute cost for LTD is too high since the metric has to be calculated for every layer; for loss-/frequency-based metrics, generally accumulated loss or frequency is used and this accumulated metric would not be changed within the same iteration (a.k.a. one forward pass of the network). Therefore, the unchanged loss/frequency metric leads the dropped token to be the same for different layers, making the token dependency not be captured by the MHA of middle layers (Vig & Belinkov, 2019).

To satisfy the independent requirement of LTD, we propose to use *purely random* token dropping assignment. For each transformer layer, we randomly (uniformly) select a small batch of tokens to proceed with the compute and drop the rest. In more details, assume $M_i = \{m_i(1), m_i(2), ..., m_i(s)\}$ is a random shuffle of $S = \{1, 2, ..., s\}$. Then the dropped token set is $J_i = \{m_i(1), m_i(2), ..., m_i(a_i)\}$ for the input of $L_{i+1}$.

**Random and Layerwise Token Dropping.** Combining layerwise token dropping with random token dropping, we have our final random and layerwise token dropping method (random-LTD), which can efficiently apply token dropping for each individual layer and can capture the attention dependency of each token with other others in middle layers with high probability.

The illustration of the comparison between standard baseline training and random-LTD is shown in Fig. 1 (an additional comparison with (Hou et al., 2022) in Fig. B.1). The pseudo-code is given in Fig. 2. For each layer, as compared to the baseline, random-LTD randomly selects (function "gather" in Fig. 2) a subset of the tokens and feeds (function "Layer" in Fig. 2) them into the transformer layer. Afterward, we combine (function "combine" in Fig. 2) the output of transformer layer with the dropped tokens to recover the full sequence length. Thus, the next layer still receives the full sequence and can repeat this process.

Since the dropped tokens of each layer are independent, there is no need for random-LTD to treat special tokens (e.g., [MASK], [CLS], [SEP], [PADDING]) differently from other normal tokens, which can further reduce the cost of computing the dropping criterion. Meanwhile, we show that special token treatment does not bring extra benefits for random-LTD on BERT pretraining in Section 5.2.

## 3.2 DROPPING SCHEDULE OF RANDOM-LTD

**Layers without Token Dropping.** While TokenBypass (Hou et al., 2022) needs to keep half of the layers in full sequence length training, random-LTD has no such limitation. Thanks to the attention-capture feature of random-LTD, we can apply random-LTD to most of the transformer layers except the first and last transformer layers.

Keeping the first and last layers in full sequence length training usually leads to better performance since (1) the first layer directly connects to the embedding, and it can help refine the raw feature; (2) the last layer directly connects to the final prediction; a feature realignment for all tokens can improve the model quality. We also provide a detailed study to show the importance of keeping the first and last layers without token dropping in Section 5.3.

**Monotonic Sequence Length Growth.** In order to reduce the gradient variance introduced by random-LTD for better training, we monotonically increase the kept sequence length throughout training (referred to as MSLG) with a linear schedule. Particularly, the dropped token set $J_i$ for the $i$-th layer gradually shrinks and the kept token set $K_i$ gradually grows as the training proceeds. Denote the size of $J_i$ ($K_i$) at step $t$ is $a_{i,t}$ ($b_{i,t}$), its final size is $0$ ($s$), and the total training iterations is $T$. Assume we want to gradually reduce the size of $J_i$ to zero at iteration $T'$ and the decreasing strength is $s_{dec}$. Then the decreasing step size is $T_{dec} = T'/(a_{0,t}/s_{dec})$, i.e., for every $T_{dec}$ iterations, the size of $J_i$ ($K_i$) reduces (increases) by $s_{dec}$. Please see Fig. 3 for an illustration of $K_i$ on GPT pretraining. We also show that MSLG outperforms the constant drop schedule with similar compute savings in Section 5.4.

## 3.3 NEW LEARNING RATE SCHEDULE FOR PRETRAINING

When performing pretraining on language models, we oftentimes use a decaying learning rate schedule based on iteration with a warmup period. Particularly, at the first few thousand or hundred iterations, warming up the learning rate is critical for distributed pretraining tasks due to its instability (Goyal et al., 2017; Li et al., 2021). However, an iteration-based schedule is not optimal for random-LTD.

First, random-LTD reduces the effective batch size of middle layers at the initial warmup phase. The effective training tokens for dropped token layers become much smaller than the baseline training. Second, for most of our training cases, MSLG does not reach the full length until $> 2/3$ of training iterations for large compute saving. At such time, the iteration-based learning rate is considerable small. And this small learning rate cannot provide efficient training dynamics for random-LTD. Therefore, to stabilize the initial training phase and to have a large enough learning rate in the later training phase, we need to increase the warmup iterations and slow down the learning rate decay. Here, we propose a new learning rate schedule based on the layerwise tokens consumption, called layer-token learning rate (LayerToken LR). Please see Appendix C for the formal and detailed description of LayerToken LR.

We emphasize that one can always tune the learning rate schedule by increasing the maximum learning rate or the warmup iterations. However, it would require a lot of engineering effort. Therefore, we propose this LayerToken LR schedule, which is more suitable for our random-LTD than the standard one. We also include a detailed comparison between the standard learning rate schedule and LayerToken LR in Section 5.5.

## 4 MAIN RESULTS

In this section, we first provide the results of random-LTD for pretraining on GPT and BERT models. We then extend random-LTD on the computer vision domain to demonstrate its broader applications. Similar to Section 3.3 and Appendix C, we use the LayerToken compute the cost to measure the total

training budget.[2] We also provide the real training time saving for GPT and BERT pretraining. Kindly note that the real-time saving depends on various factors, e.g., the implementation and hardware.

## 4.1 GPT PRETRAINING

We train GPT-3-style models with 350 million parameters (GPT-$3_{350M}$) and 1.3 billion parameters (GPT-$3_{1.3B}$) on PILE dataset (Gao et al., 2020) and the total number of training tokens is 300 billion. For random-LTD, the initial dropped token length for all middle layers is 1920 (i.e., 128 tokens are kept for compute), and it decreases by 16 for every 1.75B training tokens. After 210B training tokens, random-LTD degrades to standard training procedure with full sequence length. Theoretically, this can save 1/3 of the LayerToken training budget. See Appendix D.1 for more details.

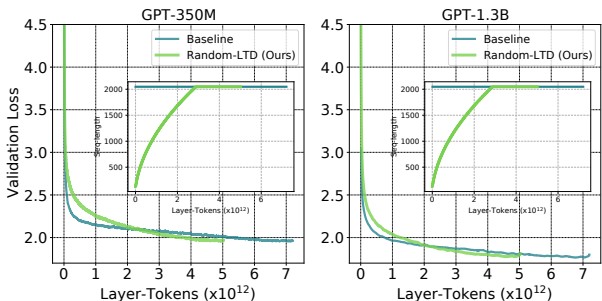

Figure 3: The comparison of validation curves between baseline and random-LTD on GPT pretraining. Here the x-axis is based on LayerToken consumption. The sequence length is illustrated in the inserted figure.

The evaluation loss curves for baseline and random-LTD are shown in Fig. 3. As can be seen, for both GPT-$3_{350M}$ and GPT-$3_{1.3B}$, random-LTD has similar evaluation losses as the baseline with 1/3 less LayerToken consumption. We also provide the zero-shot evaluation results in Tab. 1. For both GPT-$3_{350M}$ and GPT-$3_{1.3B}$, random-LTD achieves comparable results as the baseline, Besides, random-LTD can save 14.3% wall-clock training time on GPT-$3_{350M}$ and 25.6% wall-clock training time on GPT-$3_{1.3B}$.

We reiterate that the LayerToken consumption saving ratio cannot directly transfer to GPU wall-clock training time saving ratio due to the implementation/hardware. Meanwhile, note that the saving number we reported here is not the maximum potential saving (with fixed implementation/hardware, etc) since we can reduce the training GPU numbers for random-LTD at the initial training phase, which has a shorter effective training sequence length. Also, although random-LTD has the same theoretical compute saving for both GPT-$3_{350M}$ and GPT-$3_{1.3B}$, the real wall-clock time saving varies a lot because GPT-$3_{1.3B}$ has a larger hidden dim size, which means the model spends more time on real computing than other operators, e.g., data movement and gradient communication.

Table 1: Zero-shot evaluation results (last col.) on GPT-$3_{350M}$ and GPT-$3_{1.3B}$. The GPU cost provided here is the used number of A100-40G times the training days. See Tab. I.1 for 19 tasks.

| Model | Method | LayerToken Saving | GPU Cost (saving) | Ave. |
|---|---|---|---|---|
| GPT-$3_{350M}$ | baseline | None | 64×2.59 (0.0%) | 38.6 |
| | random-LTD | 33.3% | 64×2.22 (14.3%) | 38.9 |
| GPT-$3_{1.3B}$ | Baseline | None | 64×5.42 (0.0%) | 42.7 |
| | random-LTD | 33.3% | 64×4.03 (25.6%) | 42.5 |

Table 2: Finetuning results for BERT$_{large}$. The training cost provided here is the used number of A100-40G times the training days. See Tab. E.1 for full results.

| Method | LayerToken Saving | GPU Cost (Saving) | Ave. |
|---|---|---|---|
| baseline | None | 64×5.89 (0.0%) | 85.42 |
| random-LTD-1 | 26.2% | 64×5.42 (7.96%) | 86.95 |
| random-LTD-2 | 31.1% | 64×5.21 (11.5%) | 86.42 |

## 4.2 BERT PRETRAINING

We pretrain BERT$_{large}$ on PILE dataset for 2M iterations with batch size 1024 and sequence length 512 following (Shoeybi et al., 2019). We apply random-LTD with two variants, random-LTD-1 and random-LTD-2. Particularly, for random-LTD-1 (random-LTD-2), the initial kept token length is 200 (128), and it increases by 16 for every 48B (38B) training tokens. As such, we save 26.2% (31.1%) LayerToken consumption for random-LTD-1 (random-LTD-2). We evaluate the trained model on

---

[2]Similar to (Hou et al., 2022), we do not include (1) the final prediction layer and (2) the attention compute difference between the different lengths of sequence for the compute cost comparison.

four downstream tasks as (Shoeybi et al., 2019), i.e., MNLI, QQP, RACE-m, RACE-h. Note that we apply standard finetuning without token dropping to have a fair comparison for both pretrained models from the baseline and random-LTD. Please see Appendix D.2 for more details.

Tab. 2 summarizes the results along with the full results in Tab. E.1. Although random-LTD is slightly worse than baseline on a certain task (QQP, see Tab. E.1), it gives much higher accuracy on other tasks while saving 26.2–31.1% of the theoretical computation overhead in pretraining. Overall, random-LTD-1 achieves 1.54 points higher average accuracy over baseline and random-LTD-2 achieves 1 point higher average accuracy over baseline.

Meanwhile, random-LTD-1 (random-LTD-2) saves about 7.95% (11.5%) wall-clock time as compared to baseline. Note that similar to GPT pretraining, the saving depends on the implementation/hardware, and random-LTD has potentially larger savings if elastic training is performed. Also, although GPT-$3_{350M}$ and BERT$_{large}$ have similar model sizes as well as similar theoretical compute saving, the final wall-clock training time saving varies by about 3%. This is caused by BERT$_{large}$ having a shorter final sequence length (i.e, 512) than GPT (i.e, 2048), and the real compute time for a sentence with sequence length 128 is not 1/4 (or 1/16) of a sentence with 512 (2048) tokens. This leads the overall compute time-saving for GPT-$3_{350M}$ to be larger than that for BERT$_{large}$.

## 4.3 ViT Finetuning

We perform the vision transformer (ViT) on both ImageNet (with a 12-layer pretrained ViT) and CIFAR (with a 24-layer pretrained ViT). For random-LTD, the initial sequence length is 66 and linearly reaches the 197 full sequence length at 80% of the total training iterations such that 22.3% layer-token saving is achieved. See training details in Appendix D.3. We summarize the result with standard deviation in Tab. 3 along with the full details in Tab. H.1. As can be seen, random-LTD can achieve comparable results as the baseline on all three datasets. This demonstrates the broader applications of random-LTD.

Table 3: Finetuning result of ViT on ImageNet. See Tab. H.1 for the results on CIFAR10/100.

| Method | ImageNet datasets on 12-layer ViT | | |
| | LayerToken Saving | Top-1 | Top-5 |
| --- | --- | --- | --- |
| baseline | N/A | 84.65±0.04 | 97.41±0.02 |
| random-LTD | 22.3% | 84.70±0.04 | 97.48±0.02 |

Table 4: Ablation study of special token treatment for BERT pretraining with 22.2% LayerToken saving. See Tab. H.2 for all results.

| Keep Special Tokens | Pretraining PPL val/test | Downstream Ave. |
| --- | --- | --- |
| yes | 6.024 / 6.049 | 88.50 |
| no | 6.018 / 6.040 | 88.52 |

## 5 Discussion

In this section, we present several import ablation studies and the potential regularization effect of random-LTD. Besides the three tasks used in previous sections, we also include GPT finetuning on causal-LM problems using the GPT-$2_{350M}$ from Huggingface (Wolf et al., 2019). Please see Appendix D.4 for the training details. Also, we reduce the iterations of BERT pretraining from 2M to 200k due to resource limitations. Please see Appendix D.2 for more details.

### 5.1 Layerwise Token Dropping vs. TokenBypass

Although TokenBypass (Hou et al., 2022) demonstrates its great ability on BERT pretraining, its skipping policy may still hurt the performance of other tasks, e.g., causal-LM. The reason is mentioned in Section 3, i.e., MHA of middle layers focuses on different tokens at different depths. Fully skipping those layers may lead the causal-LM task to lose attention capability. However, random-LTD does not have this issue as it randomly selects kept tokens for each layer.

To verify this, we provide an ablation study on the comparison between random-LTD and TokenBypass with GPT-$2_{350M}$ finetuning on PTB (Marcus et al., 1993). We make two sets of experiments:

- **Set 1.** Following (Hou et al., 2022), we bypass half of the tokens based on their empirically moving average loss from $L_{12}$ to $L_{23}$. Similarly, we apply *constant* random token drop to the layers from the middle to the second last ($L_{12}$ to $L_{23}$).
- **Set 2.** We apply TokenBypass or constant random token dropping for half of the tokens starting from the second layer ($L_2$) until the second last layer ($L_{23}$).

The validation curves of two cases are shown in Fig. 4. As can be seen, for both cases, random-LTD performs much better than TokenBypass. Particularly, for the Set 2 comparison, the perplexity of random-LTD is about 10 points lower than TokenBypass, demonstrating that random-LTD can be applied to more layers than TokenBypass.

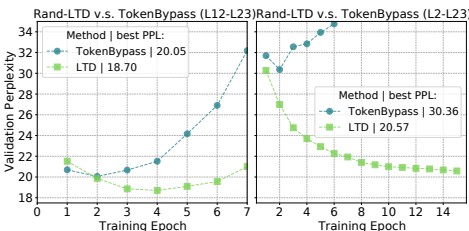

Figure 4: Comparison between random token dropping and TokenBypass.

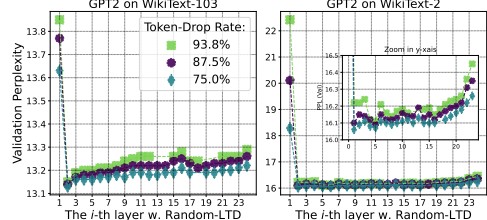

Figure 5: Sensitivity of random token dropping of each layer.

## 5.2 WITH/WITHOUT SPECIAL TOKEN TREATMENT

Different from the GPT pretraining task, which has consecutive sentences/paragraphs, the BERT pretraining data consists of two sentences that could be unrelated. The special tokens [CLS] and [SEP] play critical roles for the model in determining the beginning/end of each sentence so that the model can predict the relationship between the two sentences. Thus, there could be potential gain in keeping the tokens for all layers, which has a more detailed discussion in (Hou et al., 2022).

Here we present an ablation study on whether keeping those special tokens helps random-LTD or not. We perform a straight comparison for random-LTD: (1) one with purely random selection and (2) the other with an additional criterion, i.e., keeping the special tokens for all layers. See training details in Appendix D.2

The results of MNLI/QQP are shown in Tab. 4 along with the full details in Tab. H.5. As can be seen, for both pretraining loss and downstream finetuning, special token treatment does not provide any benefit for random-LTD. Note that random-LTD without special token treatment is also more compute-friendly for token dropping.

## 5.3 WHY WE NEED TO KEEP THE FIRST AND LAST LAYER?

To understand why keeping the first and last layers in full sequence length training, we present single layer sensitivity analysis shown in Fig. 5 for GPT-$2_{350M}$ finetuning on Wikitext-2 and Wikitext-103. Particularly, we apply constant token dropping for one layer and keep all the other layers in the standard training mode. After the training, we measure the PPL and use it as the sensitivity metric, i.e., higher PPL indicates high sensitivity and vice versa. The U-shape of both curves implies that the first and last ones are most sensitive to token dropping.

Table 5: Comparison of applying random-LTD to different layers on GPT-$2_{350M}$ finetuning and ViT finetuning. See Tab. H.3 with standard deviation.

| Metric | dataset | Apply random-LTD except for Layer | | | |
|---|---|---|---|---|---|
| | | None | First | Last | First&Last |
| Perplexity | PTB | 16.00 | 16.01 | 16.09 | 15.92 |
| | WikiText-2 | 17.06 | 17.01 | 17.01 | 16.94 |
| | WikiText-103 | 13.27 | 13.03 | 13.23 | 12.99 |
| Accuracy | ImageNet-Top1 | 84.47 | 84.51 | 84.65 | 84.70 |

Table 6: Compare between MSLG and constant token dropping schedules. See Tab. H.4 for the full result with standard deviation and the result on ViT finetuning.

| dataset | PTB (Metric: perplexity) | | |
|---|---|---|---|
| Token-drop schedules | constant | constant | MSLG |
| LayerToken saving | 23.0% | 32.1% | 33.7% |
| Performance | 18.27 | 20.76 | 15.92 |

To further understand if random-LTD can be applied to all layers when using MSLG, we include other three scenarios, i.e., applying random-LTD to (1) all but not last layer, (2) all but first layer, and (3) all layers. We perform finetuning tasks on both causal-LM and image classification. See the full training details in Appendix D.3 and Appendix D.4. From Tab. 5 and H.3, we can clearly see that keeping the first and the last layers intact leads to a substantial improvement (beyond standard deviation) over the rest three scenarios.

### 5.4 WHY WE NEED SEQUENCE LENGTH GROWTH?

We now give an ablation study on why MSLG schedule is necessary. Again, We perform finetuning tasks on both causal-LM and image classification.

We specially set a constant token dropping rate that matches the token saving of MSLG schedules with all other hyperparameters fixed. We present the results in Tab. 6 and H.4. It can be clearly seen that given the almost same amount of LayerToken saving ($33\% - 35\%$), the constant dropping schedule has worse performance than MSLG. MSLG schedule can actually be even better or comparable to those constant ones whose saving is $10\%$ smaller.

### 5.5 LAYERTOKEN LR SCHEDULE EFFECT

To study the effectiveness of LayerToken LR, we compare three training scenarios for GPT-$3_{350M}$ with 300B training tokens (see Appendix D.1 for training details): (1) the baseline training with the standard learning rate, (2) random-LTD with the standard learning rate, and (3) random-LTD with LayerToken LR. The validation curves and their corresponding learning rates with respect to iterations are plotted in Fig. 6. As can be seen, the green curve (random-LTD with LayerToken LR) can achieve comparable validation loss as the baseline, which is better than random-LTD with the standard learning rate. This confirms that the small learning rate introduced by the standard learning rate schedule slows the learning of random-LTD at the later training phase. A similar observation is made for the BERT pretraining, which will be deferred to Appendix F due to the space limit.

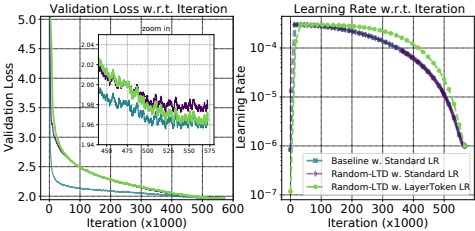

Figure 6: Comparison between a standard learning rate and our proposed learning rate based on layerwise tokens

Figure 7: Study the regularization of random-LTD on GPT-$2_{350M}$ finetuning on the PTB task without dropout (left) and with dropout (right).

### 5.6 COMPARISON ON GPT-$3_{1.3B}$ WITH VARIOUS TRAINING BUDGETS

In this section, we perform various training budgets to train GPT-$3_{1.3B}$ and compare the performance of baseline and random-LTD to verify if random-LTD can consistently save the training cost.

To fully understand the effectiveness of random-LTD, we train GPT-$3_{1.3B}$ using a baseline with 120B (i.e., 2880 billion LayerToken consumption) to 360B tokens (i.e., 8640 billion LayerToken consumption). For random-LTD, we follow our Section 4 setting to save 1/3 of the total training budget and apply it to three training budgets, which are equivalent to 2880, 4800B, and 5760B LayerTokens.

The results are summarized in Tab. 7. The first noticeable result here is that the total amount of training LayerTokens directly affects the model's final quality. Longer training usually leads to better accuracy. Meanwhile, random-LTD can use 2/3 of the training budget to achieve similar evaluation results as the baseline. For example, random-LTD with training budgets 2880, 4800, and 5760 billion LayerTokens can lead to similar accuracy as the baseline with training budgets 4320, 7200, and 8640 billion LayerTokens.

### 5.7 THE INTERPLAY BETWEEN RANDOM-LTD AND DROPOUT

Token dropping can be viewed as a special (coarse-grained) case of dropout. As such, it may have the potential to work as dropout for certain tasks or further help dropout for certain tasks. To investigate this, we apply random-LTD with/without standard dropout on both pretraining and finetuning tasks.

**BERT$_{large}$ Pretraining.** Please see Appendix D.2 for training details. For random-LTD-3 (random-LTD-4), the initial kept token length for all middle layers is 256, and it increases by 16 for every 3.8 (6) billion training tokens such that we eventually save 14.1% (22.3%) LayerToken consumption.

The results are summarized in Tab. 8 with details in Tab. H.5 and we include the pretraining perplexity to better understand the dropout effect. Clearly, turning off dropout results in lower evaluation/test perplexity (shown in Tab. 8). Meanwhile, the performance of those no-dropout models (baseline*, random-LTD-3*, random-LTD-4*) on MNLI and QQP show an obvious improvement over their dropout counterparts (baseline, random-LTD-3, random-LTD-4). However, for RACE finetuning, there is no learning for the no-dropout baseline model, which somewhat is surprising but shows the importance of dropout for pretraining. In contrast, when turning off the dropout for random-LTD, we see a compelling better accuracy on RACE, which exceeds the standard baseline pretraining by >1% on RACE-m. Thus, random-LTD brings not only the efficiency but also the potential regularization effect to BERT pertaining.

Table 7: Zero-shot results on GPT-3$_{1.3B}$ with various budgets (by Layer-Token). See Tab. I.2 for all 19 tasks.

| Method | Budget (B) | Ave. |
|---|---|---|
| Baseline | 2880 | 41.0 |
| | 4320 | 41.7 |
| | 5760 | 42.5 |
| | 7200 | 42.7 |
| | 8640 | 43.1 |
| random-LTD | 2880 | 42.1 |
| | 4800 | 42.5 |
| | 5760 | 43.1 |

Table 8: Study the regularization effect of random-LTD. We report the average of dev and test for RACE-m and RACE-h. * means no dropout. Please see Tab. H.5 for the full result with standard deviation.

| (Layer-token saving) | Pretraining PPL val/test | Downstream Ave. |
|---|---|---|
| baseline (None) | 5.78/5.80 | 80.14 |
| baseline* (None) | 5.45/5.46 | 56.76 |
| random-LTD-3 (14.1%) | 6.37/6.40 | 78.72 |
| random-LTD-3* (14.1%) | 5.79/5.80 | 81.01 |
| random-LTD-4 (22.3% ) | 6.52/6.58 | 78.16 |
| random-LTD-4* (22.3%) | 6.02/6.04 | 80.49 |

**GPT-2$_{350M}$ Finetuning.** Let us further study the potential regularization effect of random-LTD on GPT-2$_{350M}$ finetuning tasks. We train two sets of models, one without dropout and one with dropout (the default rate is 0.1). We also apply random-LTD with three initial sequence lengths— 128, 256, and 512—such that they reach the full sequence 1024 at the same epoch (12). We present their validation curves in Fig. 7.

We see that the baseline training without dropout quickly overfits, as shown in the left of Fig. 7 (black curve). And its best validation PPL is worse than the baseline with dropout (block curve in right figure). However, for all three cases of random-LTD, they achieve similar PPL as the standard baseline with dropout. Even with the default dropout in the right of Fig. 7, its validation still faces a non-negligible overfitting issue after the middle of training. In contrast, random-LTD with MSLG introduces a potential (extra) regularization effect. As can be seen, the validation curves of random-LTD are flattening and they maintain in a considerably small value regime towards the end of training.

**Summary.** We do not claim that random-LTD can replace dropout. In fact, the two can work well (note that both are turned on for the results in GPT pretraining in Section 4) with each other as dropout and random-LTD focus on different granularity. We see that random-LTD with dropout achieves lower perplexity for the small finetuning task (shown in the right of Fig. 7). Thus, the additional potential regularization effects introduced by random-LTD could be complementary to dropout and play a helpful role in those small-dataset tasks with critical overfitting issues. Also, we see that random-LTD without dropout achieves better downstream finetuning accuracy for low-budget BERT$_{large}$ pretraining. This may indicate random-LTD can be a potential regularization method for low-budget pretraining.

## 6 CONCLUSIONS

In this work, we propose a novel random and layerwise token dropping algorithm (random-LTD) along with dropping schedules and a new learning rate schedule. We demonstrate the efficiency of random-LTD on both GPT and BERT pretraining problems as well as GPT and ViT finetuning tasks. In addition, we probe all ablation studies to verify the effectiveness of each single algorithm component used in random-LTD. Furthermore, we also show the potential regularization effect introduced by random-LTD. For the discussion on the limitations and future work, see Appendix G.

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

# A  OTHER EFFICIENT TRAINING APPROACHES

Mixed-precision training (Micikevicius et al., 2017), different parallelism schemes (Shoeybi et al., 2019; Huang et al., 2019), and memory-efficient system design (Rasley et al., 2020; Rajbhandari et al., 2020) are the most commonly used system-related efficient training methods to train large-scale transformer models.

Besides those system-level optimizations, researchers and practitioners also investigate various efficient training methods. Gong et al. (2019); Li et al. (2020) propose layer stacking to speed up BERT training by gradually increasing the number of layers. Zhang & He (2020); Liu et al. (2021b) extend this idea by adaptively changing the depth of the training. Following this, Shen et al. (2022) further incorporates the growth of the width along with depth for language modelings. Rae et al. (2021) tile the weights from a small model to a larger one to reduce the training cost for larger models. Li et al. (2021) introduce curriculum learning to stabilize the training and get faster convergence of GPT models. Liu et al. (2021a) propose auxiliary self-supervised task to enable ViT to be effectively trained on small datasets.

Despite the remarkable success, those methods usually only demonstrate their capability on one specific application and do not show their generalizability across the wide usage of transformer models. In contrast, we extensively test random-LTD on both pretraining and finetuning.

# B  COMPARE BETWEEN BASELINE, TOKENBYPASS AND RANDOM-LTD

We include the illustration of the comparison between baseline, TokenBypass, and random-LTD in Fig. B.1.

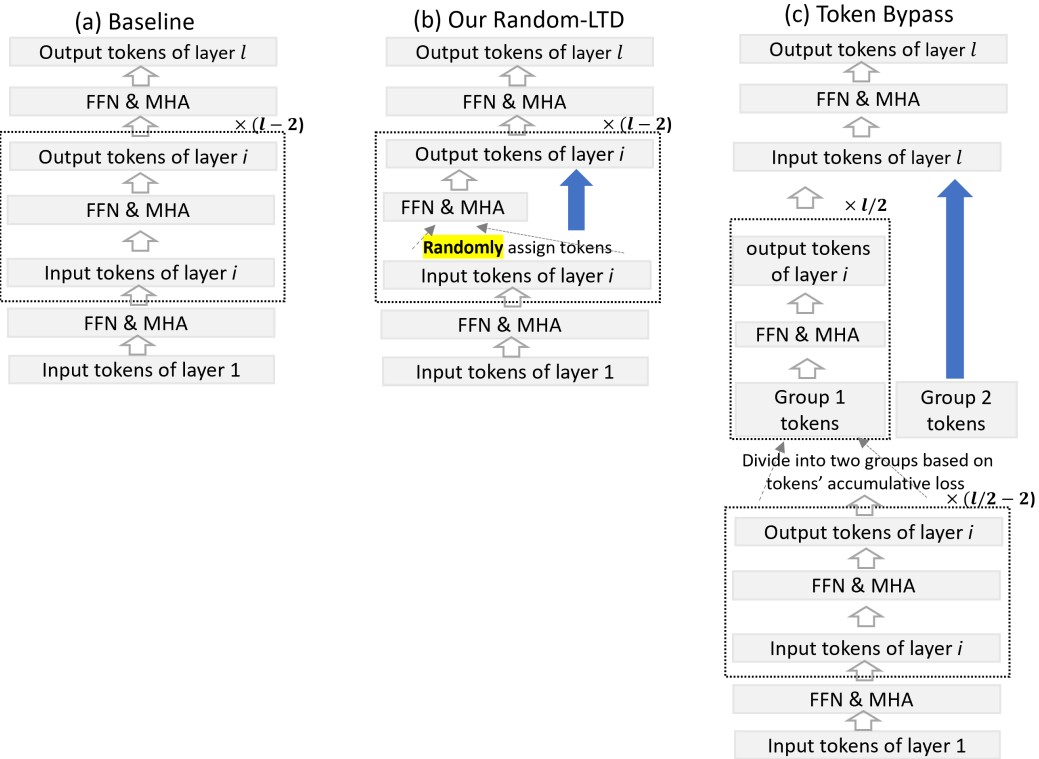

Figure B.1: Illustration of the transformer model for the baseline training (left), TokenBypass training (right) and random-LTD training (middle). Compared to TokenBypass, random-LTD requires no criterion on the dropped tokens and trains well for all middle layers. The box with dash line is a repeated block. For both (a) and (b), the block is repeated by $l - 2$ times, while for (c), the block is repeated by $l/2$. In the box, "Output tokens of layer $i$" is the same as "Input tokens of layer $i + 1$".

## C    FORMAL LAYERTOKEN LR DESCRIPTION

Formally, recall that the number of drop or kept tokens is $a_{i,t}$ or $b_{i,t}$ for layer $L_i$ at iteration $t$. Since we have the same size of $J_i$ and $K_i$ for all layers, we drop the subscript $i$ for simplicity. At iteration $t$, the total LayerToken consumed by the entire network is $2s + (l-2)b_t$ (random-LTD is applied to all middle layers). Then the total consumed tokens is $2sT + \sum_{t=0}^{T}(l-2)b_t$.

Suppose for the iteration-based, the warmup iterations is $T_{\text{warmup}}$, which means the consumed tokens for warmup in all layers is $slT_{\text{warmup}}$. Then, the warmup iterations using our LayerToken LR would be $T_{\text{LTwarmup}}$ such that

$$2sT_{\text{LTwarmup}} + \sum_{t=0}^{T_{\text{LTwarmup}}}(l-2)b_t = slT_{\text{warmup}}. \tag{1}$$

Similarly, for an iteration-based learning rate schedule, the decay learning schedule (e.g., linear decay or cosine decay schedules) is based on the length of $T - T_{\text{warmup}}$. For our LayerToken LR schedule, the corresponding length is

$$2sT + \sum_{t=0}^{T}(l-2)b_t - 2sT_{\text{LTwarmup}} - \sum_{t=0}^{T_{\text{LTwarmup}}}(l-2)b_t. \tag{2}$$

Note that this proposed schedule would reduce to standard learning rate if we keep $b_t = s$ for all layers (without random-LTD). Please see Section 5.5 for the effectiveness of our LayerToken LR as compared to standard learning rate schedule and see Fig. 6 for the learning rate schedule illustration.

## D    DETAILED EXPERIMENTAL SETUP

### D.1    EXPERIMENTAL SETUP FOR GPT PRETRAINING

We use DeepSpeed (Rasley et al., 2020) and Megatron-DeepSpeed (Megatron-DeepSpeed) repository to train GPT models with 350 million parameters (GPT-3$_{350M}$) and 1.3 billion parameters (GPT-3$_{1.3B}$). The pretraining data are from PILE dataset Gao et al. (2020) and the total training tokens are 300 billion without extra explanation. For random-LTD, the initial dropped token length for all middle layers is 1920 (i.e., 128 tokens are kept for compute), and it decreases by 16 for every 1.75 billion training tokens. That is to say, after 210B training tokens, random-LTD degrades to standard training procedure with full sequence length. Theoretically, this can save 1/3 of the layer-token training budget. All models are trained with 64 A100-40G GPUs.

We evaluate our results on 19 zero-shot evaluation tasks, including 19 accuracy evaluation tasks (i.e., HellaSwag (Zellers et al., 2019), LAMBADA (Paperno et al., 2016), TriviaQA (Joshi et al., 2017), WebQS (Berant et al., 2013), Winogrande (Sakaguchi et al., 2020), PIQA (Tata & Patel, 2003), ARC (Challenge/Easy) (Boratko et al., 2018), ANLI (R1/R2/R3) (Williams et al., 2020), OpenBookQA (Mihaylov et al., 2018), RACE-h (Lai et al., 2017), BoolQ (Clark et al., 2019), Copa (Afshar et al., 2018), RTE (Dagan et al., 2013), WSC (Levesque et al., 2012), MultiRC (Yadav et al., 2019), and ReCoRD (Zhang et al., 2018)).

### D.2    EXPERIMENTAL SETUP FOR BERT PRETRAINING

Similar to GPT pretraining, we use the Megatron-DeepSpeed repository to train our BERT$_{\text{large}}$ models with 336 million parameters (24 layers) with sequence length 512. The pretraining recipe follows (Shoeybi et al., 2019), but the pretraining data is public, the same as GPT pretraining.

For the main results in Section 4.2, the total training tokens is $512 \times 1024 \times 2 \times 10^6$. Here 1024 is the global batch size and 512 is the sequence length. We pretrain with 2 million iterations. We trained on 64 A100-40g GPUs (16 batch/GPU). For random-LTD-2 (random-LTD-1), the initial kept token length for all middle layers is 128 (200) for compute, and it increases by 16 for every 38 (48) billion training tokens such that we eventually save 31.1% (26.23%) layer-tokens. We take the last checkpoint to perform downstream tasks on the popular stable benchmarks, including MNLI (Williams et al., 2017), QQP (Iyer et al., 2017), and RACE (middle and high difficulty) (Lai et al., 2017), of which the fine-tuning configurations (same as (Shoeybi et al., 2019)) respectively are

10, 12 and 3 epochs with batch-size 128, 128 and 32 and learning rate 1e-5, 5e-5 and 2e-5. We report the median (best) of five repeated runs (random seeds 1234-1238) in Tab. 2. Note that we apply the standard training (no random-LTD) in the downstream tasks in order to have fair comparisons.

For the ablation study of BERT pretraining in Section 5.2 and 5.7, we shorten the pretraining iteration to 0.2 million (due to limited resource) and set the maximum learning rate to be 4e-4. We start with the initial sequence 256 and increase it by 16 for every 6 billion training tokens such that we eventually save 22.3% layer-tokens. To compensate for the short-time pretraining, we make the standard fine-training epoch much longer (30 epochs) for MNLI, QQP, and RACE and their batch-size (learning rate) are 128 (5e-5), 128 (5e-5), and 64 (2e-5) respectively.

### D.3 EXPERIMENTAL SETUP FOR ViT FINE-TUNING

We apply random-LTD to the vision transformer (ViT) (Dosovitskiy et al., 2021) on fine-tuning tasks in order to demonstrate the broader applications of our method across different domains. We use the pretrained models published in (Wightman, 2019) and focus mainly on the two small image recognition benchmarks— CIFAR10 and CIFAR100 (Krizhevsky et al., 2009), and one large-scale dataset—ImageNet (Deng et al., 2009). For ImageNet (CIFAR10/100), we use the 12-layer (24-layer) pretrained ViT with an input resolution $224 \times 224$ in which each patch of size $16 \times 16$ such that the sequence length becomes $196+1$ (the extra token is for position). ImageNet (CIFAR10/100) is trained on an 8-GPU (1-GPU) A100-40G machine such that the batch size is 32 (128) images per GPU. The training budget for all three datasets is 14 epochs and a small constant learning rate is used based on grid search. Particularly, the best learning rate for ImageNet is 5e-5 and CIFAR10/CIFAR100 is 1e-4. For ImageNet (CIFAR), the sequence length is started with 66 (32) and linearly reaches to the 197 full sequence length at 80% of the total training iterations such that 22.3%(30.9%) layer-token saving is achieved.

### D.4 EXPERIMENTAL SETUP FOR GPT FINETUNING

For the language fine-tuning tasks, we directly take the existing pretrained GPT model (350M, 24-layer) published in HuggingFace (Wolf et al., 2019) and fine-tune on the three dataset: Penn Treebank (PTB) (Marcus et al., 1993), WikiText-2 and WikiText-103 (Merity et al., 2017). These fine-tuning tasks are trained on a single V100 GPU with a batch size of 32 and a constant learning rate 5e-5. We trained for 15 epochs for PTB, 4 epochs for WikiText-103 and 10 epcoh WikiText-2. As for random-LTD in Tab. 5, the sequence length is started with 128 (64) sequence with a linear increase to the 1024 full sequence length at 80% (70%) of the total training iterations for PTB (WikiText-103/-2).

## E STANDARD DEVIATION FOR BERT$_{\text{LARGE}}$ DOWNSTREAM TASKS

We show the standard deviation for BERT$_{\text{large}}$ downstream tasks finetuning in Tab. E.1.

Table E.1: The comparison between baseline and random-LTD on finetuning for BERT$_{\text{large}}$. Complementary to Tab. 2, we report the mean and one standard deviation over five independent runs with the same hyperparameters.

| Method | MNLI-m/-mm | QQP | RACE-m (dev) | RACE-m (test) | RACE-h (dev) | RACE-h (test) |
|---|---|---|---|---|---|---|
| Baseline | 89.09±0.05/89.52±0.21 | 92.29±0.12 | 84.58±0.25 | 82.88±0.57 | 80.54±0.29 | 79.01±0.45 |
| random-LTD-1 | 89.73±0.09/89.93±0.15 | 92.09±0.08 | 85.67±0.32 | 84.51±0.58 | 82.78±0.28 | 81.47±0.43 |
| random-LTD-2 | 89.36±0.1/89.74±0.12 | 91.96±0.11 | 86.14±0.29 | 84.96±0.27 | 82.08±0.19 | 80.74±0.16 |

## F BERT$_{\text{LARGE}}$ PRETRAINING WITH/WITHOUT LAYERTOKEN LR

In Section 5.5, we have seen the benefits of using LayerToken LR over the standard learning rate for GPT pretraining. Here we present the additional results for BERT$_{\text{large}}$ pretraining. The training budget is 0.2 million iterations, and the training details are given in Appendix D.2. The results are presented in Tab. F.1. which further confirms the dominant benefit of LayerToken LR.

Table H.3: Comparison of applying random-LTD to different layers on GPT-2$_{350M}$ finetuning and ViT finetuning. This is the full result of Tab. 5.

| Method | dataset | Random-LTD applied to all layers except for the following | | | |
|---|---|---|---|---|---|
| | | None | First | Last | First and Last |
| Perplexity | PTB | 16.00±0.02 | 16.01±0.03 | 16.09±0.02 | 15.92±0.02 |
| | WikiText-2 | 17.06±0.02 | 17.01±0.02 | 17.01±0.02 | 16.94±0.01 |
| | WikiText-103 | 13.27±0.01 | 13.03±0.03 | 13.23±0.04 | 12.99±0.01 |
| Accuracy | ImageNet-Top1 | 84.47±0.08 | 84.51±0.08 | 84.65±0.04 | 84.70±0.04 |

Table F.1: Results of BERT$_{large}$ pretraining with 0.2 million iterations for random-LTD with 14.1% layer-token saving. Random-LTD-3* in Tab. 8 is the same as the last row (LayerToken LR).

| Learning rate method | Pretraining results | | DownStream finetuning results | | | | | |
|---|---|---|---|---|---|---|---|---|
| | ppl(val) | ppl(test) | MNLI-m/-mm | QQP | RACE-m (dev) | RACE-m (test) | RACE-h (dev) | RACE-h (test) |
| Standard LR | 5.90 | 5.92 | 86.46/86.53 | 91.82 | 72.09 | 73.79 | 73.50 | 71.04 |
| LayerToken LR | 5.79 | 5.80 | 87.14/87.29 | 91.95 | 78.05 | 77.84 | 73.47 | 71.30 |

## G  LIMITATIONS AND FUTURE WORK

We believe it is critical for every work to clearly state its limitations, especially in this area. An important limitation of this work is that we keep the dropped token ratio for all intermediate layers the same. We design it in such a way as to reduce the parameter tuning effort. However, each layer may have its sensitivity (see Fig. 5). Therefore, an automated dropped ratio could help here. Another limitation is that our MSLG is based on a linear increasing manner. This might not be optimal, and a self-adaptive schedule could further improve the efficiency and convergence behavior. Finally, in this work, we found out random-LTD can have the potential regularization effect as the standard dropout. However, to improve its generalizability, there are still a lot of experiments to be done, which is out of the scope of this paper. We leave this as future work.

## H  FULL RESULTS USED IN MAIN TEXT

We include the full results used in main text in this section.

Table H.1: Finetuning result of ViT on ImageNet and CIFAR. This is the full result of Tab. 3.

| | ImageNet datasets on 12-layer ViT | | | CIFAR datasets on 24-layer ViT | | |
|---|---|---|---|---|---|---|
| | LayerToken Saving | Top-1 | Top-5 | LayerToken Saving | Top-1 (CIFAR100) | Top-1 (CIFAR10) |
| baseline | N/A | 84.65±0.04 | 97.41±0.02 | N/A | 93.93±0.30 | 99.32±0.05 |
| random-LTD | 22.3% | 84.70±0.04 | 97.48±0.02 | 30.9% | 94.02±0.40 | 99.30±0.03 |

Table H.2: Ablation study of special token treatment for BERT pretraining with 22.2% LayerToken saving. This is the full result of Tab. 4.

| Keep Special Tokens | Pretraining results | | DownStream finetuning results | |
|---|---|---|---|---|
| | ppl(val) | ppl(test) | MNLI-m/-mm | QQP |
| yes | 6.024 | 6.049 | 86.70/86.97 | 91.83 |
| no | 6.018 | 6.040 | 86.66/86.92 | 91.97 |

## I  FULL ZERO-SHOT EVALUATION OF GPT-STYLE MODELS

We include all zero-shot evaluation results for all GPT models in Tab. I.1 and I.2.

Table H.4: Compare between MSLG and constant token dropping schedules. This is the full result of Tab. 6.

| datasets | CIFAR10 (Metric: Top-1 accuracy) | | | | PTB (Metric: perplexity) | | |
|---|---|---|---|---|---|---|---|
| Token-drop schedules | constant | constant | constant | MSLG | constant | constant | MSLG |
| LayerToken saving | 16.5% | 23.6% | 30.8% | 32.3% | 23.0% | 32.1% | 33.7% |
| Performance | 99.33±0.01 | 99.28±0.01 | 99.26±0.08 | 99.32±0.03 | 18.27±0.08 | 20.76±0.06 | 15.92±0.02 |

Table H.5: Study the regularization effect of random-LTD. We report the average of dev and test for RACE-m and RARCE-h. * means no dropout.

| Method | Pretraining results | | DownStream finetuning results | | | | | |
|---|---|---|---|---|---|---|---|---|
| (Layer-token saving) | ppl(val) | ppl(test) | MNLI-m/-mm | QQP | RACE-m (dev) | RACE-m (test) | RACE-h (dev) | RACE-h (test) |
| baseline (None) | 5.78 | 5.80 | 86.44/86.51 | 92.11 | 75.85 | 75.50 | 73.78 | 70.78 |
| baseline* (None) | 5.45 | 5.46 | 86.4/86.93 | 92.07 | 32.67 | 32.46 | 34.52 | 32.26 |
| random-LTD-3 (14.1%) | 6.37 | 6.40 | 85.91/85.97 | 91.84 | 74.72 | 73.22 | 70.95 | 68.43 |
| random-LTD-3* (14.1%) | 5.79 | 5.80 | 87.14/87.29 | 91.95 | 78.05 | 77.84 | 73.47 | 71.30 |
| random-LTD-4 (22.3% ) | 6.52 | 6.58 | 85.69/85.64 | 91.74 | 73.65 | 71.88 | 70.49 | 68.00 |
| random-LTD-4* (22.3%) | 6.02 | 6.04 | 86.66/86.92 | 91.97 | 76.78 | 76.56 | 73.38 | 71.15 |

Table I.1: Zero-shot evaluation results of baseline and random-LTD on GPT-3$_{350M}$ and GPT-3$_{1.3B}$.

| Tasks | Baseline | | random-LTD | |
|---|---|---|---|---|
| | GPT-3$_{350M}$ | GPT-3$_{1.3B}$ | GPT-3$_{350M}$ | GPT-3$_{1.3B}$ |
| HellaSwag | 39.3 | 52.1 | 40.2 | 51.8 |
| LAMBADA | 52.3 | 61.2 | 52.3 | 62.2 |
| TriviaQA | 3.6 | 6.3 | 3.14 | 6.05 |
| WebQs | 1.82 | 2.21 | 1.58 | 1.92 |
| Winogrande | 53.3 | 55.7 | 50.9 | 58.0 |
| PIQA | 67.0 | 71.1 | 67.2 | 71.0 |
| ARC (Challenge) | 25.2 | 29.4 | 24.9 | 28.2 |
| ARC (Easy) | 46.0 | 53.4 | 44.4 | 53.2 |
| ANLI R1 | 32.7 | 32.9 | 33.1 | 33.5 |
| ANLI R2 | 32.6 | 33.7 | 33.0 | 32.4 |
| ANLI R3 | 33.8 | 35.1 | 33.7 | 34.6 |
| OpenBookQA | 28.8 | 33.4 | 29.6 | 32.6 |
| RACE-h | 29.3 | 33.4 | 31.8 | 34.2 |
| BoolQ | 55.6 | 56.4 | 58.4 | 62.7 |
| Copa | 67.0 | 71.0 | 68.0 | 72.0 |
| RTE | 51.6 | 56.7 | 53.1 | 52.7 |
| WSC | 36.5 | 43.3 | 36.5 | 36.5 |
| MultiRC | 0.84 | 0.84 | 0.84 | 0.84 |
| ReCoRD | 76.3 | 82.3 | 76.4 | 82.9 |
| Average Acc | 38.6 | 42.7 | 38.9 | 42.5 |

Table I.2: Zero-shot evaluation results of baseline and random-LTD on GPT-$3_{1.3B}$ with various training budgets. Here, "Budget" in the first column means the training layer-token as the final budget.

| Tasks/Budgets (B) | Baseline | | | | | random-LTD | | |
|---|---|---|---|---|---|---|---|---|
| | 2880 | 4320 | 5760 | 7200 | 8640 | 2880 | 4800 | 5760 |
| HellaSwag | 46.4 | 49.2 | 51.9 | 52.1 | 52.9 | 49.1 | 51.8 | 53.4 |
| LAMBADA | 57.1 | 59.0 | 60.2 | 61.2 | 61.5 | 60.2 | 62.2 | 62.8 |
| TriviaQA | 5.36 | 5.58 | 7.73 | 6.3 | 7.2 | 4.79 | 6.05 | 7.46 |
| WebQs | 2.31 | 2.02 | 1.33 | 2.21 | 2.26 | 1.97 | 1.92 | 1.72 |
| Winogrande | 54.7 | 55.1 | 57.5 | 55.7 | 56.6 | 56.4 | 58.0 | 60.1 |
| PIQA | 70.1 | 70.1 | 71.1 | 71.1 | 72.3 | 70.5 | 71.0 | 70.7 |
| ARC (Challenge) | 26.0 | 26.6 | 27.8 | 29.4 | 28.8 | 26.2 | 28.2 | 28.9 |
| ARC (Easy) | 51.1 | 52.0 | 52.5 | 53.4 | 54.6 | 51.6 | 53.2 | 53.2 |
| ANLI R1 | 33.0 | 31.3 | 32.7 | 32.9 | 32.1 | 33.4 | 33.5 | 33.2 |
| ANLI R2 | 32.6 | 32.2 | 35.7 | 33.7 | 33.4 | 32.7 | 32.4 | 33.0 |
| ANLI R3 | 35.2 | 33.8 | 36.9 | 35.1 | 35.8 | 34.8 | 34.6 | 32.9 |
| OpenBookQA | 31.2 | 31.2 | 33.0 | 33.4 | 33.6 | 33.6 | 32.6 | 34.2 |
| RACE-h | 32.8 | 34.4 | 34.0 | 33.4 | 36.6 | 34.3 | 34.2 | 35.7 |
| BoolQ | 62.0 | 58.9 | 61.5 | 56.4 | 59.9 | 61.7 | 62.7 | 62.8 |
| Copa | 71.0 | 72.0 | 70.0 | 71.0 | 74.0 | 73.0 | 72.0 | 71.0 |
| RTE | 52.0 | 58.1 | 54.2 | 56.7 | 56.3 | 56.7 | 52.7 | 56.3 |
| WSC | 36.5 | 36.5 | 36.5 | 43.3 | 36.5 | 36.5 | 36.5 | 37.5 |
| MultiRC | 0.94 | 2.62 | 0.94 | 0.84 | 1.26 | 0.84 | 0.84 | 0.84 |
| ReCoRD | 79.5 | 81.2 | 82.2 | 82.3 | 83.2 | 81.6 | 82.9 | 83.4 |
| Average Acc | 41.0 | 41.7 | 42.5 | 42.7 | 43.1 | 42.1 | 42.5 | 43.1 |

