# OpenReview forum: "Efficient Large-scale Transformer Training via Random and Layerwise Token Dropping"
_ICLR.cc/2023/Conference — Submitted to ICLR 2023_

### Official Review · Reviewer_YEPB · 2022-10-23

**Confidence:** 4
**Correctness:** 2
**Technical Novelty And Significance:** 1
**Empirical Novelty And Significance:** 2
**Recommendation:** 3

**Clarity, Quality, Novelty And Reproducibility:**

The basic description of the methods is quite clear.  However, other parts of the paper were less clear.  In particular, I found it hard to understand the reasons why random-LTD would be expected to outperform TokenBypass on GPT, and some of the material on the experiments seemed to list large amounts of detail while sometimes not making it explicitly clear what the key question being answered by the experiment was, and why the experimental setup was justified.

Regarding novelty, the methods are essentially a small tweak on TokenBypass; the experimental measurements could be valuable but the methodological novelty is very limited.

Reproducibility of the experiments seems adequate to me.

**Strength And Weaknesses:**

Strengths

The paper's method is simple to understand, and the paper includes many experiments.  The methods are appropriately described in the context of the most similar prior methods, and both the proposed method and the prior ones are described clearly.  In the experiments, the proposed method does offer meaningful improvements over standard training, and unlike previous work this paper shows that it is effective on both BERT and GPT.  The ablations isolate interesting aspects of the design.

Weaknesses

The most significant weakness in this paper is that the experiments do not adequately demonstrate the value of the technique over previous work.  We don’t see a comparison against TokenBypass on any end tasks, which given that the paper is claiming that TokenBypass performs poorly on GPT and the proposed method performs better, is a serious limitation.  The only comparison against TokenBypass on GPT is in a very limited PennTreebank perplexity evaluation, where the best TokenBypass achieves 1.3 points worse perplexity than the submission’s method.  We don’t see a comparison on end tasks with BERT or GPT pretraining on TokenBypass.  Finally, the pretraining experiments appear to involve only a single run of the proposed system and the baseline, so it is somewhat unclear how reliable the conclusions are.

The paper also claims that its method requires far less manual design effort than TokenBypass, but I was somewhat unconvinced about this claim.  While it's true that the paper's method does not require importance weights, rules for special tokens, or the “sandwich rule”, I'm not sure these differences are that significant.  This paper has its own somewhat complex design details (as the ablations show): e.g., the fraction of dropped tokens linearly decreases as training proceeds, which is important for performance.  Likewise, the authors have to invent an associated learning rate that accounts for the number of retained tokens.


**Summary Of The Paper:**

This paper presents random-LTD (Layer Token Dropping), a method that randomly skips the computation of certain tokens during pretraining, and evaluates it on BERT and GPT pretraining, and ViT fine-tuning.  The paper includes several ablations to show that the design decisions behind its method are necessary.

**Summary Of The Review:**

This paper presents experiments showing that a token dropping method is effective, however despite being fairly extensive the experiments leave do not adequately support the claims of random-LTD's superiority to TokenBypass.  The novelty of the proposed method is also quite limited.

---

> ### Author Response · Authors · 2022-11-16
> **Response to Reviewer YEPB [Part I]**
>
> **Q1**: We don’t see a comparison against TokenBypass on any end tasks, which given that the paper is claiming that TokenBypass performs poorly on GPT, and the proposed method performs better, is a serious limitation.
>
> **Answer**: Please see our general reply which includes more comparisons between our method and TokenBypass. Also, for end tasks, we are not sure what this means, and we assume it is about finetuning tasks. However, as pointed out in the general reply, TokenBypass is not easy to be extended for classification/regression tasks, as well as popular Q&A benchmark, e.g., Squad, since those tasks do not have accumulated token loss. We hope this addresses your concerns.
>
> ----------------------------------
>
> **Q2**: The pretraining experiments appear to involve only a single run of the proposed system and the baseline, so it is somewhat unclear how reliable the conclusions are.
>
> **Answer**: Note that for pretraining tasks, the computation cost is considerable large (hundreds of thousands of GPU hours). It’s by default and acceptable in academia to use one random seed such as the well-known pretrained models roBERTa[1] and GPT3[2], the efficient pretraining methods [3,4] and even for those pretraining knowledge distilled models [5]. Also note that we provide multi-random seed runs for fine-tuning tasks.
>
> [1] Liu, Yinhan, et al. "Roberta: A robustly optimized bert pretraining approach." arXiv preprint arXiv:1907.11692 (2019).
>
> [2] Brown, Tom, et al. "Language models are few-shot learners." Advances in neural information processing systems 33 (2020): 1877-1901.
>
> [3] Shen, Sheng, et al. "Staged Training for Transformer Language Models." arXiv preprint arXiv:2203.06211 (2022).
>
> [4] Li, Conglong, Minjia Zhang, and Yuxiong He. "Curriculum learning: A regularization method for efficient and stable billion-scale gpt model pre-training." arXiv preprint arXiv:2108.06084 (2021).
>
> [5] Jiao, Xiaoqi, et al. "Tinybert: Distilling bert for natural language understanding." arXiv preprint arXiv:1909.10351 (2019).
>
> -----------------------------
>
> **Q3**: This paper has its own somewhat complex design details (as the ablations show): e.g., the fraction of dropped tokens linearly decreases as training proceeds, which is important for performance. Likewise, the authors must invent an associated learning rate that accounts for the number of retained tokens.
>
> **Answer**: We believe the linear sequence schedule and the Layer token learning rate are not complicated for practitioners to apply and use as we have shown its abroad application without much hyperparameter tuning.
>
> Particularly, for linear sequence schedule increasing, the overall tuning hyperparameter is about how much cost you want to save. Based on the cost, the schedule can be quickly generated. A similar thing happens to TokenBypass (the saving ratio along with the skipped layers) and a lot of other training saving methods as well, i.e., how much saving you want to apply [4]. Also note that, in our general reply (Table 2), with various saving ratios (up to 40%), random-LTD is able to achieve comparable performance as baseline. This also shows the robustness of our method.
>
> In addition, we do not believe that LayerToken learning rate schedule is a complicated design. First of all, when there is no random-LTD, LayerToken schedule degrades to standard learning rate schedule. Second, it is a plug-and-play replacement module as nowadays learning rate schedule, which does not require any tuning (at least based on our experiments).
>
> We hope this answered your questions

---

> > ### Author Response · Authors · 2022-11-16
> > **Response to Reviewer YEPB [Part II]**
> >
> >
> > **Q4**: (a) While it's true that the paper's method does not require importance weights, rules for special tokens, or the “sandwich rule”, I'm not sure these differences are that significant. (b) Regarding novelty, the methods are essentially a small tweak on TokenBypass; the experimental measurements could be valuable, but the methodological novelty is very limited.
> >
> > **Answer**:  We kindly pointed out that the “complex design” in Q3 (although we disagree the two components are complex) and the multiple simplifications in question (a), i.e., random-LTD does not require importance weights, rules for special tokens, or the sandwich rule, as compared to TokenBypass, are already answered the reviewer’s question (b), i.e., the novelty is limited.
> >
> > Besides, we want to clarify that a simpler method does not mean there is no novelty and/or contributions. For instance, one work before ResNet is called Highway Network (https://arxiv.org/pdf/1505.00387.pdf), which used a weighted sum to combine the results from the residual map. However, ResNet simplifies this by using an identical map, which provides better accuracy and wider applications (nowadays, the majority of works use identical maps instead of the weighted sum).
> >
> > Similarly, comparing random-LTD with TokenBypass, we simplify a lot of criteria used in TokenBypass as well as extend the applications by introducing new components. Note that we did not claim our work is as great as ResNet, but we want to emphasize that simplification does not mean the work has limited novelty and/or contributions.

---

### Official Review · Reviewer_kbJK · 2022-10-24

**Confidence:** 4
**Correctness:** 3
**Technical Novelty And Significance:** 3
**Empirical Novelty And Significance:** 2
**Recommendation:** 5

**Clarity, Quality, Novelty And Reproducibility:**

- Clarity

Overall, the paper is well-written and easy to understand. However, there are some parts that are hard to understand due to the lack of explanations (see Suggestions above).

- Quality

The quality of the experiments is impressive. They conduct extensive experiments and ablation studies in a range of experimental settings. However, more discussions on the method (e.g., how and why it works even with the random policy? or more comparisons against TokenBypass) should be further made.

- Novelty

The idea is not entirely novel. However, they empirically show that such a simple algorithm works well on both BERT and GPT pretraining. It is clear that this method has not yet undergone thorough research.

- Reproducibility

I cannot find the reproducibility statement in the main manuscript. The authors do not provide any codes or software to reproduce their results. However, the authors do provide some details on the implementations and experimental setups in the Appendix.

**Strength And Weaknesses:**

### Strengths

- Simple but effective idea; The idea is really simple. The authors empirically verify their simple algorithm works well on both BERT and GPT pre-training.
- Thorough analysis and ablation studies; This work conduct thorough analysis and ablation studies on their proposed method. It is beneficial because practitioners may gain wide insights and confidence in its feasibility from such extensive experiments.

### Weaknesses

- Limited significance and novelty; This paper has practical contributions to the field since the authors extensively validate their method. However, the idea of token dropping during training is already proposed in a previous work TokenBypass [1]. Although the authors already discussed and compared against TokenBypass [1] a lot in the main paper, they conduct limited comparison against TokenBypass [1] only in  GPT-2 (Figure 4). I think it is better to include the comparison also in the BERT-like model.
- Limited Analysis on “why” it works; The authors empirically demonstrate that pretraining with random-LTD significantly lowers pretraining's computational costs while maintaining the language model's task performance. But does it apply to all models and architectural sizes? Large language models used in the experiments by the authors include BERT-large and GPT-3 1.3B. There is no proof, yet, that this approach can scale up to much larger models like the GPT-3 135B. Then, how do practitioners choose to pre-train their huge language models using this method? I don't recommend that the authors do experiments on all language model sizes; rather, I believe they should provide some explanations (or at least conjectures) for why and how the random token drop method is effective.
Consider dropping tokens depending on their token (or word) frequency as an alternative to dropping them at random. It is assumed that the (random) token dropping works well since it eliminates some types of duplicate training during the pre-training stage if such a technique (e.g., token frequency-based dropping) outperforms random dropping.

### Suggestions

- Questions
    - What is the LayerToken? I fail to find any definition on LayerToken. As far as I understand, LayerToken indicates the sum of the number of tokens used in every layer. Am I right? If not, please clarify this term.
    - What is the baseline in Figures 3, 6, 7, and Tables 1, 2, 7, 8? Does it indicate the pre-training without any token dropping?
- Typos

    page4: we use the LayerToken ‘compute’ the cost to measure the total → to compute?

    page8: Again, ‘We’ perform → Again, we perform


[1] Hou et al., Token Dropping for Efficient BERT Pretraining, ACL 2022

**Summary Of The Paper:**

This paper proposes the idea of random layerwise token dropping method. Extensive experimental restuls empirically demostrate that randomly dropping tokens during pre-training preserves the task accuracy but saves the computational cost for pre-training large language models.

**Summary Of The Review:**

This paper does have technical contributions, considering its extensive and thorough experiments on layerwise random token dropping in both BERT and GPT pretraining.

I am not sure about the significance of this work. This work may be beneficial to some practitioners who want to pretrain large language models with lower costs. However, I think more extensive analysis and discussions beyond ablation studies are further required.

Considering its practical contributions, I rate this paper as a weak rejection.

---

> ### Author Response · Authors · 2022-11-16
> **Response to Reviewer kbJK**
>
> We thank the reviewer’s assessment and questions.
>
> **Q1**: Although the authors already discussed and compared TokenBypass [1] a lot in the main paper, they conduct limited comparison against TokenBypass [1] only in GPT-2 (Figure 4). I think it is better to include the comparison also in the BERT-like model.
>
>
> **Answer**:  Please see our general replies. Particularly, we emphasize that our method is more easily applicable to BERT finetuning as we need no criteria of dropping, while it’s not easy to extend TokenBypass to finetuning task as there is no accumulated MLM loss (per token loss) for classification problems.
>
>
> ---------------------------------
>
> **Q2**: There is no proof, yet, that this approach can scale up too much larger models like the GPT-3 135B.   I believe they should provide some explanations (or at least conjectures) for why and how the random token drop method is effective.
>
> **Answer**: This is a great point. We want to clarify that due to the huge cost, most of efficient training works, such as "Curriculum learning: A regularization method for efficient and stable billion-scale GPT model pre-training" and “Staged Training for Transformer Language Models”, have not been able to approve their ability for extra-large transformer models (like the >100B as the reviewer pointed out). However, they still provide good insight of how to train models efficiently.
>
> Here we provide more intuition and explanations why we expect our method to work (or benefit) for extra-large models. We will also add it to our final revision.
>
> The key components that make random-LTD work are as follows: (1) the layer-wise random selection helps the attention mechanism capture the dependency between different tokens. As such it does not fully lose the attention capability in the middle layers of each token with other tokens (See Section 5.1 in the paper and results in Table 2 in our general reply); (2) The gradual sequence length growth method also makes the final stage of random-LTD have the same training sequence as baseline method (see results in Section 5.4 in the paper and the comparison of results of TokenBypass in Table 1 and 2 in our general reply). Both features won’t change as the model size increases.
>
> Meanwhile, the entire procedure of random-LTD can be viewed as “coarse-to-fine-grained" training. That is to say, each layer randomly selects learning parts/tokens at the beginning, and they eventually learn the full sequence.  This procedure helps the training have smoother training. Particularly, Figure 3 in the papers shows that
> (1) baseline usually has a sharp loss decreasing at the beginning (like 1/7 of the full training) and then has a very flat loss curve (the rest 6/7);  (2) random-LTD has a much longer but slow initial decreasing phase (until 2/3 of the training).
>
> This demonstrated that at the beginning phase, there is no need to feed the full sequence for all layers. Also, note that the cross point between baseline and random-LTD happens around the time when random-LTD has the full sequence, which means the slow training at the beginning can be recovered as the sequence length increases.
>
> We hope this answer addressed your concerns.
>
> --------------------------
>
> **Q3**:
> (a) What is the LayerToken? I failed to find any definition on LayerToken. As far as I understand, LayerToken indicates the sum of the number of tokens used in every layer. Am I right? If not, please clarify this term.
> (b) What is the baseline in Figures 3, 6, 7, and Tables 1, 2, 7, 8? Does it indicate the pre-training without any token dropping?
>
> **Answer**:
> (a) Yes, you are right that LayerToken is the sum of the number of tokens used in every layer. The official definition is included in Appendix C and we agree it is hard to find. We will move the definition to the main text in the final revision.
> (b) yes, the baseline is always standard training without any token dropping.
>
> --------------------------
>
> **Q4**: Reproducibility
>
> **Answer**: We will open source our codes to the public with both finetuning and pretraining examples.

---

> > ### Comment · Reviewer_kbJK · 2022-11-25
> > **Response**
> >
> > Thanks to the authors for the response and I appreciate your response that clarifies why and how the random token drop method can be effective.
> >
> > After reading the authors' response and other reviews, I still lean forward to the weak rejection.
> >
> > I agree that this work provides enough empirical evidence on the effectiveness of random layerwise token dropping for large-scale transformer pre-training. The practitioner may be able to reduce their computational budgets by referencing the provided experimental results to train their own extensive pre-trained model.
> >
> > However, I also agree with other reviewers' concerns about the novelty of this work, since their proposed method (random layerwise token dropping) seems a bit trivial. Therefore, I am not sure whether this work has enough contribution to be accepted at this conference.

---

### Official Review · Reviewer_1ZdE · 2022-10-25

**Confidence:** 4
**Correctness:** 3
**Technical Novelty And Significance:** 2
**Empirical Novelty And Significance:** 2
**Recommendation:** 5

**Clarity, Quality, Novelty And Reproducibility:**

The quality of the paper is valid as it clearly describes the methodology and conduct multiple evaluations on its proposed method under different scenarios. However, it lacks competitive baseline methods to justify the performance of the proposed method. For details, see `Strength and Weakness` section.

This paper contains all the details of experiment setting such as number of runs and hyper-parameter configurations, which provides helpful information for other researchers to reproduce the work.

**Strength And Weaknesses:**

Strength:
1. This paper is well-written and clearly describes a novel methodology for token pruning that can be used for pre-training and fine-tuning.
2. It evaluates the method in several scenarios of both pre-training and finetuning using GPT-3 and BERT for both NLP and vision domain.
3. Moreover, it gives various ablation studies such as comparing random-LTD with SOTA TokenBypass and investigating if LTD-random requires special token treatment.

Weakness:
1. Insufficient baseline methods: this paper describes his main novelty in comparison to TokenBypass method (Hou et al., 2022). However, it only compares its random-LTD with a baseline model without any token pruning in the main results and the comparison with TokenBypass in the ablation studies is insufficient. In particular, it would be interesting to see the comparison results in the following setting. (a) At the same (or similar) level of computational cost, how random-LTD is performed with TokenBypass (in terms of accuracy) in both pre-training and finetuning with GPT and BERT. (b) try different values of computational cost (i.e., different values of kept token length), at each value, how random-LTD is performed with TokenBypass. It is expected that, as the computational cost becomes smaller (more tokens are pruned), accuracy drops from the baseline model (which does not have any token pruning) becomes larger. But it will be good to see that random-LTD has smaller accuracy drop from the baseline model without any token pruning, in comparison to the TokenBypass method.

2. Lack of justification on using random method to select tokens. The token pruning method consists of three major parts: (1) how many token to prune at each layer (which is a essentially hyper-parameter in the random-LTD), (2) given the number of tokens to prune, which tokens are kept and which ones are pruned (i.e., token selection method), and (3) mechanism to recover full sequence length that will benefit pre-training and other fine-tuning task that require full sequence length for final prediction. It is understandable that the proposed random-LTD cannot be directly compared with inference only pruning methods such as Pyramid-BERT or PowerBERT. However, given (1) and (3) being fixed, we can compare token selection method between random method with other token selection methods such as the coreset based token selection method in Pyramid-BERT (https://arxiv.org/abs/2203.14380) or attention score based token selection method in PowerBERT (https://arxiv.org/abs/2001.08950). I.e., under different values of computation cost, replace the random method in random-LTD by coreset and attention score based token selection method.

Also, another naive baseline is given the number of tokens to prune k, truncating the first k tokens and see how it compares with the random method. Because of the dummy tokens appended at the end of each sequence to make the uniform input sequence length, truncating the first k tokens sometimes can give better performance as it simply just removes the unimportant dummy tokens.

Some questions:
1. Will the `combine` function in figure 2 preserve the original sequence order?
2. For GLUE tasks, is the embeddings of CLS token on top of the transformer used for final classification or the pooling of embeddings from all tokens in the top layer?
3. For random selection method, will random-LTD conduct random selection based on real tokens only or based on all tokens including dummy ones? It would be good to have those details in the paper.




**Summary Of The Paper:**

This paper proposes a novel random and layer-wise token dropping method (random-LTD), which skips the computation of a subset of the input tokens at all middle layers. The new random-LTD method does not require any importance score-based metrics but just random selection, and hence saves computational cost. In addition, it does not require many layers in full sequence length training except the first and the last layers. This method is evaluated in several scenarios of both pre-training and finetuning using GPT-3 and BERT.

**Summary Of The Review:**

This paper proposes a novel random and layer-wise token dropping method (random-LTD), which skips the computation of a subset of the input tokens at all middle layers. The new random-LTD method does not require any importance score-based metrics but just random selection, and hence saves computational cost. In addition, it does not require many layers in full sequence length training except the first and the last layers. This method is evaluated in several scenarios of both pre-training and finetuning using GPT-3 and BERT. However, it lacks competitive baseline methods such as TokenBypass and different token selection methods (coreset based token selection) in the experimental section to justify the performance of the proposed method.

---

> ### Author Response · Authors · 2022-11-16
> **Response to Reviewer 1ZdE (Part I)**
>
> We thank the reviewer’s assessment and questions.
>
> **Q1**: Insufficient baseline methods, the comparison with TokenBypass in the ablation studies is insufficient.
>
> **Answer**: Please see our answer in the general reply.  We have performed both GPT pretraining and fine-tuning experiments. We hope our experiments addressed your questions.
>
> **Q2**: Lack of justification on using random method to select tokens.  (1) how many tokens to prune at each layer (2) given the number of tokens to prune, which tokens are kept (3) mechanism to recover full sequence length. Given (1) and (3) being fixed, we can compare token selection method between random method with other token selection methods in Pyramid-BERT or  PowerBERT. I.e., under different values of computation cost, replace the random method in random-LTD by coreset and attention score-based token selection method.
>
> **Answer**: We thank your suggestions on varying dropping criterion (2) while fixing (1) and (3). One of the biggest advantages of our random selection criterion is that it is very cheap to apply so that the great improvement in the pretraining efficiency can be obtained. Using the metrics suggested by Pyramid-BERT, PowerBERT and TokenBypass would inevitably bring additional computation costs and make the method more complicated/difficult to apply on a lot of tasks, particularly on pretraining tasks.
>
> We here use PowerBERT as an example. First of all, Since PowerBERT is designed for downstream task inference speedup, the PowerBERT has a decreasing sequence length as the inputs go to deeper layers (see Figure 1 of PowerBERT paper). However, in our case, we have a constant sequence length for all middle layers. Using deceasing sequence length cannot be applied to pretraining since we need to do token-wise prediction (note that some modification can be happened here, e.g., skip-connecting the dropped token to the final last layer for prediction. However, exploring all those possibilities is out of the scope of our paper. We also conjecture that this method might work well for BERT-pretraining but not necessary for GPT, which is similar to TokenBypass.).
>
> Therefore, we need to apply attention score-based methods (based on PowerBERT) for *layer-wise* token selection on BERT-large (the sequence is 512) or GPT (the sequence is 2048).  Then the whole pipeline of pretraining the BERT/GPT layer will be much complicated than random-LTD (Figure 2 in the main text), which looks like this (we use pytorch-like pseudo-code here):
> ```
> # Begin of one BERT/GPT layer, we use x as the input here
> # Compute Q, K with Linear layer Query and Key
> Q, K = Query(x), Key(x)
>
> # Production between Q, K for different heads with function: Split_and_prod
> Attn_score = Split_and_prod(Q, K)
>
> # Softmax to get the final attention probability
> Attn_prob = Softmax(Attn_score)
>
> # Compute attention-based scores and then sort with function: Score_and_sort
> selected_pos = Score_and_sort(Attn_prob)
>
> # Use the same gather function as random-LTD (in Figure 2, also see answer to Q1 of Reviewer JdQ2 for details about the function) to get the reduced length x_sub
> x_sub = gather(x, selected_pos)
>
> # To save compute, gather the attention map from Attn_score with a function gather_attn
> Attn_score_sub = gather_attn(Attn_score, selected_pos)
>
> # Use x_sub and Attn_score_sub for all rest compute, including Attention probability, Value, Self-attention output, and the entire FFC. Assume the final output is x_sub_final
>
> # Use the same combine function as rondom-LTD (in Figure 2, also see answer to Q1 of Reviewer JdQ2 for details about the function) to put x_sub back the full sequence
> combine(x, x_sub_final)
>
> # Next layer
> ```
> Note that the overhead of all those operators (except gather/combine since random-LTD also uses them) is not a small portion of the entire training: (1) from the compute perspective, this attention-score based method needs to compute the attention probability for a longer sequence (always 512 for BERT-large or 2048 for GPT) as compared to random-LTD (based on current sequence length). (2) The non-compute bounded operators, like "Score_and_sort", "gather_attn", and the extra "Softmax" to get attention probability, are very costly since they are memory bounded.
>
> As such, applying some metrics, e.g., attention-scored based methods for pretraining with layer-wise token selection, contradicts the objective of our paper (get real speedup). This part of the discussion is included in Section 3.1 on Page 3 (paragraph Random Token Dropping). We will highlight this discussion and add more details in the later version for better clarification.

---

> > ### Author Response · Authors · 2022-11-16
> > **Response to Reviewer 1ZdE (Part II)**
> >
> >
> > **continue answer to Q2** In the paper (Section 5.1), we compared the result of random-LTD and TokenBypass. However, Section 5.1 is not comprehensive. Therefore, in the general reply, we add more comparisons of different saving ratios on GPT finetuning, and add TokenBypass plus MSLG, as well as one run on GPT pretraining. We will include those results in the final revision.
> >
> > Also, we are happy to include a paragraph in future work about this line of research, for instance, it would be interesting to have a comprehensive comparison of various token-dropping criteria and various sequence schedules, from different aspects, like model quality and real efficiency.
> >
> > -----------------------------------
> >
> > **Q3**: another naive baseline is given the number of tokens to prune k, truncating the first k tokens and see how it compares with the random method. Because the dummy tokens appended at the end of each sequence to make the uniform input sequence length, trunca ting the first k tokens sometimes can give better performance as it simply just removes the unimportant dummy tokens.
> >
> > **Answer**: Thanks a lot for the great suggestion. We did some analysis and eventually did not apply this baseline. The reasons are as follows.
> >
> > We remark that for GPT pretraining and ViT finetuning, all sequences have the same fixed length and there are no “dummy tokens” appended at the of each sequence. As for BERT, we analyzed the training data (i.e., PILE dataset) we used in the paper and found that more than 55% of sequences are at the max length 512 without any padding.
> >
> > For BERT-large pretraining, we have an experiment on dropping the padding first and then randomly drop other tokens. The results show there is only tiny difference between this method and randomly dropping the entire sequence. The pretraining validation ppls are 5.79 (drop padding first and then random drop tokens) and 5.83 (purely randomly drop), separately. However, the “drop padding first then others” also introduces extra cost as the padding length for each sequence is different and we need to select tokens sentence-by-sentence, which slows down the speedup. Since, the main goal of this paper is to introduce a practical method for pre-training speedup, we eventually use random drop mechanism for all models, including GPT and BERT.
> >
> > -----------------------------------
> >
> > **Q4**
> > (a) Will the combine function in figure 2 preserve the original sequence order?
> >
> > (b) For GLUE tasks, is the embeddings of CLS token on top of the transformer used for final classification or the pooling of embeddings from all tokens in the top layer?
> >
> > (c)For random selection method, will random-LTD conduct random selection based on real tokens only or based on all tokens including dummy ones? It would be good to have those details in the paper.
> >
> > **Answer**:
> >  (a) This is a great point. Yes, it is needed to preserve the order, particularly for auto-regressive tasks (note that this can also be done by re-creating the attention mask, which is more costly).
> >
> > (b) We use the [CLS] token (i.e., the standard Huggingface tuning pipeline, which uses the Pooler Layer) to do the final classification.  Also, note that for the results in GLUE task shown in Table 2, we still use standard finetuning instead of random-LTD due to the short sequence length used in finetuning and its associated limited time saving.
> >
> >  (c) This is a great question. As we stated in the paper, we choose tokens purely based on a random manner. Therefore, we use both real tokens and padding tokens during random selection.

---

### Official Review · Reviewer_JdQ2 · 2022-11-02

**Confidence:** 4
**Correctness:** 4
**Technical Novelty And Significance:** 2
**Empirical Novelty And Significance:** 2
**Recommendation:** 6

**Clarity, Quality, Novelty And Reproducibility:**

The paper is generally well written. However, the terms are overly defined in some cases while some terms are not exactly defined. For example, gather and combine operations in Figure 2 should be explained with equations. I understood what the term LayerToken means but I coudn’t find the formal definition.

I likes the comprehensiveness of ablation studies on all components of the proposed method in the paper.

I am curious about the performance of models trained with random-LTD with token dropping at inference time. I believe the trained model will be robust to token dropping at inference time.


**Strength And Weaknesses:**

random-LTD achieves reasonable computation saving while keeping accuracy. For my understanding, random-LTD is more like a regularization method as the authors also mentioned though itself also gives efficiency gain during the training. It would be great if this two effects could be differentiated.


**Summary Of The Paper:**

This paper proposes an efficient method to train large-scale transformer models by random and layerwise token dropping, called random-LTD. This method skips the computation of randomly selected tokens in intermediate layers accepte the first and the last layer. The schedule of sequence length and its corresponding learning rate are used for stable and efficient training. Experiments on GPT pre-training, BERT pre-training, and ViT fine-tuning demonstrate that random-LTD achieves similar (or better) performance compared to standard training while saving computational cost about 20-30%.

**Summary Of The Review:**

Overall, the proposed method sounds reasonable and the justification of each component is verified with ablation study. The method is not difficult to implement. Considering the exhaustively expensive costs of training large transformer models, this paper is practically useful. I think the paper could benefit from the revised writing.

---

> ### Author Response · Authors · 2022-11-16
> **Response to Reviewer JdQ2**
>
> We thank the reviewer’s positive assessment and questions.
>
> **Q1**: the terms are overly defined in some cases while some terms are not exactly defined. For example, gather and combine operations in Figure 2 should be explained with equations. I understood what the term LayerToken means but I coudn’t find the formal definition.
>
> **Answer**: Sorry about the confusion. The definition of LayerToken is given in Appendix C. We will move the definition from the Appendix to the main text. We appreciate the suggestion to explain the function “gather” and “combine” with equations. We now explain more about the two functions here (and will add more equations and illustrations to the paper in the final revision).
>
> Let us take a model GPT-350M for example. The sequence length (seq) is seq=2048. Assume at iteration t, the sequence length is s(t)<seq, for instance, 128, defined by the linearly increasing schedule.
>
> The function “Gather” randomly selects 128 tokens out of 2048 for each sample. For example, the position we select is pos_token = [1, 2, …, 128] and the length of pos_token is 128. The function “combine” then puts the output of the transformer layer back to the input tensor with the same position based on pos_token. That is to say, the order of the sequence is reserved, and the order reserved feature is needed for auto-regressive tasks.
>
> --------------------
>
> **Q2**: I am curious about the performance of models trained with random-LTD with token dropping at inference time. I believe the trained model will be robust to token dropping at inference time.
>
> **Answer**: That’s a great suggestion. The model trained with our method cannot be directly used for token-dropping inference. The reason is that in order to achieve the best pretraining accuracy/quality, we include the Monotonic Sequence Length Growth (MSLG) method in our paper. As such, the late stage of random-LTD degrades to baseline (standard) training, which loses the ability to be directly applied to inference. Also, from the paper, e.g., Table 6 and H.4, and the general reply, i.e., Table 2, the MSLG is very important to boost the quality of the model.

---

> > ### Comment · Reviewer_JdQ2 · 2022-11-25
> > **Thanks for the response**
> >
> > Dear authors,
> >
> > Thank you for the author response.
> > I hope A1 will be addressed in the revision.
> >
> > Best,
> > Reviewer JdQ2

---

### Author Response · Authors · 2022-11-16
**General response to reviewers on more comparisons/study between our random-LTD and TokenBypass (Hou et al., 2022)**

**Answer**: All reviewers asked for more comparisons/studies between our random-LTD and TokenBypass (Hou et al., 2022). Particularly (1) Reviewer 1Zde requested more comparison since the current comparison is not sufficient; (2) Reviewer kbJK asked for more comparisons on BERT-like models; and (3) Reviewer YEPB asked more comparison on end tasks (We assume that end tasks mean finetuning tasks. Please correct us if our consumption is wrong.)

We here answer this specific general question.

First, the takeaway from TokenBypass can be summarized into (M1) drop unimportant tokens starting from an intermediate layer of the model,(M2) the dropping schedules is a fixed constant function (drop half of the tokens), and (M3) the dropping criterion based on the “accumulated masked language modeling loss” (which is referred to as “token loss” since it needs each token’s loss)

However, there are several limitations (1) only tested on BERT pretraining (we find that it’s less effective in GPT-pretraining and fine-tuning), (2) the bypass layer starting only from an intermediate layer (e.g., 6L for BERT-base), and (3) the dropping criterion based on “token loss” which may not be accessible for some tasks, like classification problems.

Acknowledging that we are inspired by their excellent work and trying to solve their limitations, we propose Radom-LTD consisting of three differences: (N1) drop tokens starting from the 2nd layer of the model, (N2) propose a linear increasing dropping schedule to close the training and inference discrepancy, and (N3) the new random dropping criterion (which has lower overhead and can be easily applied to tasks without “token loss”, such as vision transformer).

We perform extensive performance to show Random-LTD's broad applications to different models, particularly for both decoder (GPT-like) and encoder (BERT-like and ViT) models (TokenBypass can only work for BERT-like models). As requested by reviewers, here we provide more direct comparisons between random-LTD and TokenBypass, and give more explanations on some aspects as well.

---

> ### Author Response · Authors · 2022-11-16
> **Part 1: Fine-tuning GPT with a fixed sequence length and various saving ratios**
>
> In the paper (Section 5.1), we compare fine-tuning GPT with a fixed sequence length and one single saving ratio. Here, to better demonstrate the benefit of random selection per layer, we provide a more compressive study with various saving ratios. Particularly, from the second layer to the last second layer, we use one of the sequence lengths from the list [921, 819, 716, 614, 512, 409], of which the corresponding token saving ratio are shown in the tables. We finetuned GPT-2 (24 layers) on the PTB dataset with the constant learning rate 5e-5 and Adam optimizer for 15 epochs (batch-size 8).  The results are the best validations (average of three runs and one standard deviation) of random-LTD (without Monotonic Sequence Length Growth, MSLG) and TokenBypass.
>
> #### **Table 1** (the standard baseline is 16.11±0.04 in a unit of perplexity (ppl, the lower the better))
> |      layer-token saving                |     1.88%     |     12.75%    |     23.72%    |     34.59%    |     45.45%    |     56.43%    |
> |-----------------------------|---------------|---------------|---------------|---------------|---------------|---------------|
> |     Random-LTD (w.o. MSLG)  |   16.15±0.01  |   16.83±0.06  |   17.95±0.08  |   20.02±0.05  |   23.35±0.16  |   30.65±0.78  |
> |     TokenBypass             |   16.4±0.04   |   17.3±0.06   |   18.59±0.19  |   23.09±0.23  |   28.56±0.24  |   35.91±0.26  |
>
> As can be seen, for all cases, random-LTD has better performance than TokenBypass, even without one of the key contributions, Monotonic Sequence Length Growth (MSLG). This further verifies the conjecture we made in the paper: “However, several works (Vig & Belinkov, 2019; Michel et al., 2019; Voita et al., 2019) have shown that MHA focuses on different tokens at different layer depths and the attention map aligns with the dependency relation most strongly in the middle of transformer architectures. Therefore, TokenBypass used in Hou et al. (2022), i.e., fully skipping middle layers, may hinder the learnability/generalization of the architecture during pretraining/inference.”

---

> > ### Author Response · Authors · 2022-11-16
> > **Part 2: Combine TokenBypass with our proposed MSLG on GPT-2 Finetuning**
> >
> > Based on Reviewer 1ZdE’s suggestion, we are also curious if MSLG can help boost the performance of TokenBypass. Therefore, we also perform the comparison between random-LTD with TokenBypass (with MSLG) on GPT-2 finetuning. We start at sequence length from 128 and linearly increase to full sequence 1024, with a slope that controls the LayerToken saving shown in the tables. The rest of the hyperparameters are the same as Part 1. The results are the best validations (average of three runs and one standard deviation) of ours and theirs are given below.
> > ##### **Table 2** (the standard baseline is 16.11±0.04 in a unit of perplexity (ppl, the lower the better))
> > |    layer-token   saving                 |     8%       |     16%       |     24%       |     32%       |     40%       |     47%       |     52%       |     55%       |
> > |------------------------------|--------------|---------------|---------------|---------------|---------------|---------------|---------------|---------------|
> > |     Random-LTD               |   15.91±0    |   15.86±0.06  |   15.86±0.01  |   15.85±0.02  |   16.05±0.06  |   17.02±0.05  |   18.41±0.04  |   20.01±0.06  |
> > |     TokenBypass  (w. MSLG)    |   16.1±0.02  |   16.09±0.05  |   16.21±0.03  |   16.54±0.01  |   17.06±0.04  |   18.64±0.04  |   23.12±0.22  |   25.77±0.57
> >
> > Note that it is slightly hard to control the drop ratio to be the same. But comparing Table-2’s 24%/47%/55% of TokenBypass with Table-1’s 23.72%/45.45%/56.43%, we can clearly see the benefit of MSLG.
> >
> > Meanwhile, comparing the results of Random-LTD and TokenBypass (with MSLG), it can be seen that Random-LTD still has better performance than TokenBypass for all cases. This shows that the other components, particularly the layerwise dropping mechanism has its unique advantage over accumulated token loss for auto-regressive generative models.

---

> > > ### Author Response · Authors · 2022-11-16
> > > **Part 3: GPT Pretraining Comparison**
> > >
> > > In our current manuscript, we did not include TokenBypass experiments for GPT pretraining. The reasons are: (1) TokenBypass cannot work well on GPT finetuning and (2) GPT pretraining and finetuning are based on the same task (i.e., causal LM). Therefore, the poor performance of TokenBypass on GPT finetuning is a good indicator of its performance on GPT pretraining.
> > >
> > > Now since the reviewers ask about **more comparison on Random-LTD for pretraining**, we present a GPT-350M pretraining with 30B tokens (due to limited time/resources, we have to shorten the training token). From the Part 1 and Part 2 study, we include MSLG for TokenBypass on GPT pre-training as it provides much better performance. The results of random-LTD and TokenBypass (with MSLG) are given below in Table 3 (it’s a fixed random-seed run due to the cost). As can be seen, random-LTD has significantly better performance than TokenBypass.
> > >
> > > ###### **Table 3**: GPT-350M Pretraining with 30B tokens
> > > |  Pretraining method                     |     Baseline  |     Random-LTD  (Saving 37.76% layer-tokens)  |     TokenBypass with MSLG  (Saving 37.76% layer-tokens)  |
> > > |----------------------------------------------|---------------|-----------------------------------------------|----------------------------------------------------------|
> > > |     Validation loss  (the lower the better)  |   8.22        |   8.26                                        |   9.62                                                   |

---

> > > > ### Author Response · Authors · 2022-11-16
> > > > **Part 4: Answer to more end task comparison**
> > > >
> > > > Thanks to Reviewer YEPB’s suggestion on trying to apply TokenBypass and Random-LTD on more end tasks, we realize that TokenBypass cannot be easily extended to downstream tasks as well. The reason is that the TokenBypass criterion is based on the “token loss”, but downstream tasks, e.g., classification and regression (GLUE benchmark), do not have “token loss”. Therefore, we did not find an easy extension for TokenBypass.

---

> > > > > ### Author Response · Authors · 2022-11-16
> > > > > **Part 5: Answer to more BERT-like model comparison**
> > > > >
> > > > > Finally, regarding the lack of comparison on **pretraining BERT models**, the reason is that we believe both methods work well on BERT.  In TokenBypass (Table 4. Please note that the end-to-end token saving of Table 4 needs to be divided by 2 since it bypasses half of layers), we can see with 62.5% drop (i.e., 31.25% saving), the accuracy of TokenBypass is similar (slightly worse) than baseline. Their saving and results are comparable to our results shown in Table 2 in the main text. We would like to reiterate that we do not say that TokenBypass (Hou et al., 2022) is worse than ours on BERT pretraining. All we are saying is that we (1) greatly simplify their token dropping criterion, and (2) empirically proves that random dropping criterion can be as effective as their MLM loss criterion and can be easily applied to many more tasks beyond BERT. In fact, random dropping criterion is so effective that the token can be dropped at the second layer resulting in more computation saving for pretraining.

---

### Decision · Program_Chairs · 2023-01-20

**Decision:**

Reject

**Justification For Why Not Higher Score:**

Most reviewers complained about lack of baselines and missing a highly relevant prior work.

**Justification For Why Not Lower Score:**

N/A

**Metareview: Summary, Strengths And Weaknesses:**

This paper presents random Layer Token Dropping, a method that randomly skips the computation of certain tokens during pretraining. Experiments on GPT and BERT pre-training, and ViT fine-tuning demonstrate that similar or better performance compared to standard training while being more efficient. The reviewers found the work to be well motivated, and found to work to be valuable in saving compute while not hurting performance. However, reviewers raised several concerns regarding lack of depth in experiments including baselines and  lack of comparisons with highly relevant work.